# HC2D: Attention-Based Two-Phase Distillation for Transformer Continual Learning

## Abstract

Catastrophic forgetting refers to marked degradation in performance on previously learned tasks after training on new ones, and continual learning aims to mitigate this problem. Many existing works preserve past knowledge by constraining updates within the locally learned representations. However, such locality can hinder the discovery of genuinely novel discriminative cues, thereby intensifying the stability–plasticity dilemma. Inspired by hippocampal–cortical memory theory and the principle of introspection, we propose a novel training framework: Hippocampal-to-Cortical Two-Phase Distillation (HC2D). In Phase I (Pattern Separation), the introspective negative attention regularizer suppresses reuse of the original core peaks but preserves global directional consistency, guiding the student to discover novel, complementary discriminative cues and forming the hippocampal teacher. In Phase II (Cortical Consolidation), we selectively consolidate the hippocampal teacher's most salient attentional patterns into the cortical backbone via asymmetric distillation without compromising the primary attention distribution. HC2D leverages Vision Transformer–based attention distillation to implement an "aggressive exploration first, robust consolidation later" strategy, with virtually no additional inference overhead. Experimental results show that HC2D consistently mitigates catastrophic forgetting and becomes increasingly effective over longer task sequences, offering a biologically inspired and engineering-efficient solution for transformer-based continual learning.

## 1 Introduction

Deep neural networks typically follow a one-shot offline learning paradigm, but when the data distribution evolves over time and novel classes appear in a continual-learning setting, such static models suffer from *catastrophic forgetting*, a steep performance drop on previously learned tasks. To alleviate forgetting(Wickramasinghe et al., 2023), numerous approaches impose knowledge-distillation or weight regularization constraints that penalize parameter drift in the vicinity of the old decision boundaries. Across these methods, interference between new and old tasks is primarily addressed via local constraints around existing decision boundaries(Wang et al., 2024), thereby biasing training toward conservative adaptation near the learned representations and leaving no explicit mechanism to actively discover novel discriminative cues. The consequence is twofold: limited representational capacity for new classes and a higher risk of feature drift. The issue is aggravated in Vision Transformer (ViT) architectures (Dosovitskiy, 2020), once certain attention heads collapse or become rigid during incremental attention distillation training, the model's sensitivity to novel patterns is further restricted, thereby hindering the learning of subsequent tasks.

Motivated by the cognitive-neuroscience memory theory (McClelland et al., 1995) and the notion of introspection(Jack & Shallice, 2001), we propose a new attention-based two-phase distillation framework called *Hippocampal-to-Cortical Two-Phase Distillation (HC2D)*, explicitly tailored for ViTs to balance exploration and integration. According to the memory theory, humans form fast but fragile representations in the hippocampus when encountering novel stimuli; these high-information traces are later consolidated into the neocortex via replay(Frankland & Bontempi, 2005). Introspection complements this process by deliberately revisiting and analyzing past decisions after each new experience. Inspired by this dual mechanism, HC2D operates as follows: Phase I (Pattern Separation (Yassa & Stark, 2011)) - Starting from a frozen snapshot of the previous task, we train the student with introspective negative distillation (Pelosin et al., 2022), which suppresses the original

core attention and encourages exploration of complementary attention patterns. This yields discriminative cues that are distinct from existing representations, and the checkpoint at the end of this stage is adopted as the hippocampal teacher. Phase II (Cortical Consolidation (Foster & Wilson, 2006)) - Under the joint guidance of the hippocampal teacher and the snapshot from the previous task, we achieve attention complementarity via asymmetric attention distillation, selectively aligning the student's attention maps, enabling effective integration of new knowledge while minimizing performance degradation on prior tasks. The two phases together realize a "*explore first, consolidate later*" routine.

In summary, the proposed HC2D integrates asymmetric distillation with an introspection-driven attention mechanism that actively explores novel information. On the benchmarks we observe a significant reduction in average forgetting, with virtually unchanged inference cost—demonstrating HC2D's low forgetting, high adaptability profile over long task sequences. And its advantage widens as the task sequence length increases. The entire framework is purely data-driven, requires no biological priors and introduces no extra inference overhead, showcasing the potential of injecting biological inspiration into attention distillation for stronger continual learning.

## 2 RELATED WORK

Continual learning requires a deep model to assimilate sequentially arriving tasks or classes while simultaneously mitigating the catastrophic forgetting of previously acquired knowledge (Rebuffi et al., 2017). A diverse spectrum of architectures has been explored to tackle this challenge. With the rapid progress of Transformer models, their reach has extended far beyond natural language processing into computer vision. In particular, the Vision Transformer (ViT) partitions (Han et al., 2023) an image into a series of patch tokens and employs self-attention to achieve state-of-the-art performance on numerous vision benchmarks (Wang et al., 2022b). Consequently, researchers have begun to leverage ViT as the backbone for continual learning. Representative studies include LVT (Wang et al., 2022a), which introduces inter-task attention and a dual-classifier design, and (Douillard et al., 2021) present Dytox, which injects a vectorized Task Token for every incoming task, thereby improving performance with minimal parameter overhead.

Knowledge Distillation (KD) (Hinton et al., 2015) plays an even more pivotal role in continual learning and is now regarded as a cornerstone technique for alleviating forgetting (Hou et al., 2018). The canonical paradigm is to train the student using a composite loss that linearly combines cross-entropy and a distillation term (Gou et al., 2021). Building upon this foundation, an extensive set of variants such as multi-teacher KD(Liu et al., 2020), feature-level KD, and attention-based KD further enriches the framework(Passban et al., 2021). Among them, attention distillation is particularly influential (Wang et al., 2020). By shaping intermediate attention(Wang & Yoon, 2021), it can stabilize optimization under distribution shift, and enhance generalization in architectures where attention governs the computation(Xu et al., 2024). This forms the theoretical foundation of the framework we propose. Many studies have also employed attention distillation to strengthen continual learning: for instance, (Mohamed et al., 2023) present D3Former, a debiased dual-distilled Transformer for class-incremental learning that mitigates bias against old classes and preserves discriminative focus by distilling spatial attention maps; similarly, (Pelosin et al., 2022) highlight the limitations of symmetric losses and propose an asymmetric variant of pooled-output distillation tailored to ViTs, which motivates our asymmetric design.

Despite the notable progress of existing methods, key challenges remain in balancing the stability-plasticity dilemma and scaling to large task corpora. Motivated by the hippocampal-neocortical memory theory and the cognitive notion of introspection(Madaan et al., 2023), we introduce a two-phase distillation framework. Unlike conventional approaches, our strategy reduces forgetting and boosts accuracy for new tasks with zero additional inference cost; its advantages become increasingly pronounced as the task sequence lengthens, offering a more efficient solution for large-scale continual learning.

## 3 METHOD OVERVIEW

In this section, we present in detail the *Hippocampal-to-Cortical Two-Phase Distillation (HC2D)* framework. Class-incremental learning inherently balances two conflicting objectives: acquiring

discriminative cues for the upcoming classes while retaining knowledge gathered from prior tasks. Prevailing distillation-centric and parameter regularization techniques impose rigid constraints in the vicinity of historical decision boundaries(De Lange et al., 2021). As a result, the model's capacity to explore novel patterns is curtailed. Motivated by the hippocampal-cortical two-phase memory hypothesis and the cognitive notion of introspection, we propose HC2D, a training paradigm that explicitly separates exploration and consolidation into two complementary phases. The sequential training protocol is detailed next. Throughout we denote the memory buffer by $\mathcal{M}$. Let $\langle \mathcal{D}^{(0)}, \mathcal{D}^{(1)}, \ldots, \mathcal{D}^{(T)} \rangle$ be a chronologically ordered stream of $T + 1$ tasks. During training, the learner is restricted to the current task data $\mathcal{D}^{(t)}$ and a small replay buffer. HC2D framework is compatible with any form of storage-based strategy, and in the current implementation we adopt a fixed capacity replay buffer that keeps only a class-balanced, memory-efficient subset of past samples.

For task 0, the inaugural task is handled by conventional supervised learning: the Vision Transformer is optimized on $\mathcal{D}^{(0)}$ with cross-entropy. Naturally, the model may be trained with any conventional training recipe; such choices exert no material influence on the effectiveness of the proposed HC2D framework. For every incoming task $t, (> 0)$, HC2D organizes learning into four orderly steps—Snapshot, *Phase I (Pattern Separation)*, *Phase II (Cortical Consolidation)*, and Memory Update. We begin by elucidating the snapshot mechanism, whereby the current backbone parameters are cloned into a frozen replica prior to the arrival of any new data. This copy encapsulates the network's memory and acts as the teacher in subsequent stages. At the onset of each new task, the framework enters Phase I. We first clone the backbone snapshot from the previous step to set as student. To promote exploration, we have the student avoid the teacher's saliency peaks while maintaining global alignment; this steers the student toward complementary cues, in line with hippocampal pattern separation. After exploration of Phase I, the student is frozen and promoted to hippocampal teacher. This phase corresponds to the left panel of Figure 1, labeled Pattern Separation. In Phase II, the distillation objective now shifts from divergence to integration: the backbone tries to keep its decisions on old classes remain stable, while only the most diagnostic information are transferred. Injecting the salient information by asymmetric distillation rather than indiscriminately copying all of them can avoid propagating noise or redundancy. This phase corresponds to the right panel of Figure 1, labeled Cortical Consolidation. Finally, a class-balanced fraction of $\mathcal{D}^{(t)}$ is inserted into $\mathcal{M}$ and ready for the next task. The whole framework of HC2D forms a tightly coupled pipeline, each assuming a distinct responsibility. The exploration phase unlocks novel discriminative subspaces, while the subsequent consolidation phase converts those discoveries into durable competence. The two-phase ensure an optimal balance between plasticity and stability.

## 3.1 PHASE I: PATTERN SEPARATION

Neuroscience suggests that, in mammals, the hippocampus rapidly encodes new experiences as volatile, high-resolution traces before any cortical consolidation takes place(Yassa & Stark, 2011). Meanwhile, continual-learning systems often suffer from stability-plasticity dilemma: once a model has discovered high-response regions for past tasks, subsequent updates tend to re-use the same saliency and thus overlook emerging cues(Grossberg, 2013). Drawing on the notion of *introspection*, we propose that new model should actively re-examine the focus in order to carve out representational space for new knowledge. Transposing this principle to class-incremental learning, we emulate the hippocampal pattern-separation mechanism at the algorithmic level. Specifically, we introduce *Introspective Negative Attention Distillation* to get the model with novel representations.

In this phase, to minimize conflicts during attention fusion in Phase II, we restrict attention distillation to the CLS-to-patch attention vectors emitted by the last decoder block, update only the last Transformer block and the classifier head, and freeze all preceding layers(Touvron et al., 2021). Here, CLS denotes the classification token in Vision Transformers; the CLS-to-patch attention vector refers to the per-head attention weights from the CLS query to all patch tokens. To further promote a smooth merge at this layer, we augment the introspective negative distillation with a global cosine alignment term, preserving overall directional consistency while "avoiding peaks" and encouraging non-core uniformization. These choices curb mid-layer distribution drift and reduce conflicts and instability. At the start of task $t$ $(t > 0)$, the backbone obtained at task $t-1$ is copied and frozen as the teacher $f_\phi$ and clone as student $f_{\theta_{\text{init}}}$. We merge the current task set $\mathcal{D}^{(t)}$ with the replay buffer $\mathcal{M}^{(t-1)}$ into a mixed dataset $\mathcal{D}_{\text{mix}} = \mathcal{D}^{(t)} \cup \mathcal{M}^{(t-1)}$. Unless otherwise stated, all training objectives are computed over this dataset. This design enables the learner to explore genuinely new cues

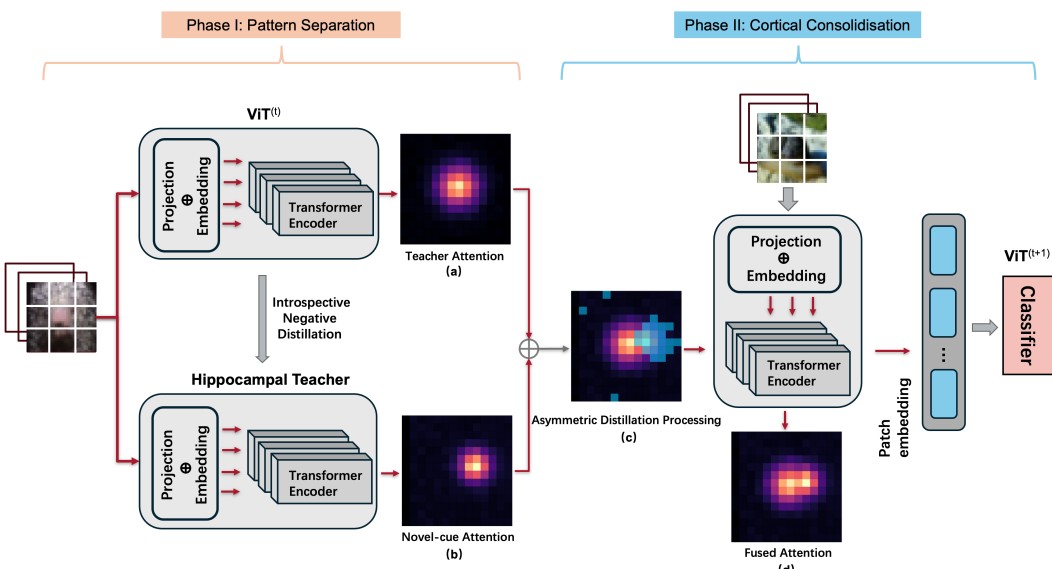

Figure 1: *Left—Pattern Separation.* The top row shows the teacher $\text{ViT}^{(t)}$ from previous task. The input image is patched and passed through *Projection $\oplus$ Embedding* and the encoder to produce the last block CLS-to-patch attention ( **a**). We use $\text{ViT}^{(t)}$ as a teacher and train the student on the same mixed data with Introspective Negative Distillation (IND), yielding the Hippocampal Teacher whose attention redistributes toward novel cues ( **b**). *Right—Cortical Consolidation.* The backbone fuses attention from the $\text{ViT}^{(t)}$ and Hippocampal Teacher via Asymmetric Distillation ($\oplus$). Panel ( **c**) visualizes the attention-fusion process, the blue-highlighted regions indicate locations whose attention is up-weighted. Panel ( **d**) denotes $\text{ViT}^{(t+1)}$ after asymmetric distillation has fused old and new knowledge, exhibiting the resulting fused attention. Overall, the figure illustrates "separate to explore" on the left and "consolidate to fuse" on the right. Note that for this figure, we intentionally use simple signals rather than concrete images as inputs, in order to make the attention operations more intuitive and easier to interpret.

with support from the current data, rather than mining exclusively from past samples. With $\mathcal{D}_{\text{mix}}$, we query the last decoder block of the frozen snapshot teacher $f_\phi$ and the trainable student $f_{\theta_{\text{init}}}$ to obtain attention vectors $a_\phi$, $a_s \in \mathbb{R}^{B \times H \times N}$, where $B$ is the batch size, $H$ the number of heads, and $N$ the number of patch tokens. Then we drop the CLS column if present and normalize along the token dimension, yielding attentions $\tilde{a}_s, \tilde{a}_\phi \in \mathbb{R}^{B \times H \times N}$.

**Introspective Negative Distillation (IND):** Given the extracted attention vectors, we further mine novel discriminative cues to refine the decision boundaries. Rather than driving unconstrained divergence between teacher and student, we impose a *selective* regularizer built from a token-level *top-mass* core of the teacher. We work with token-normalized CLS-to-patch attentions $\tilde{a}_s, \tilde{a}_\phi \in \mathbb{R}^{B \times H \times N}$. Indices $b \in [B]$, $h \in [H]$, $n \in [N]$ refer to batch, head, and token, respectively; e.g., $\tilde{a}_\phi(b, h, \cdot) \in \mathbb{R}^N$ is the token vector of sample $b$ and head $h$. For each $(b, h)$, let $\sigma_{b,h}$ sort $\tilde{a}_\phi(b, h, \cdot)$ in descending order and define the smallest prefix that covers mass $q \in (0, 1)$:

$$k^\star(b, h) = \min\left\{ k : \sum_{i=1}^{k} \tilde{a}_t(b, h, \sigma_{b,h}(i)) \geq q \right\}. \tag{1}$$

The core and non-core masks are

$$M_{\text{core}}(b, h, n) = \mathbb{I}(n \in \{\sigma_{b,h}(1), \ldots, \sigma_{b,h}(k^\star)\}), \qquad M_{\text{nc}} = \mathbf{1} - M_{\text{core}}. \tag{2}$$

The Phase I regularizer comprises three terms: (i) *core suppression* avoids reusing the teacher's peak tokens, (ii) *non-core uniformization* raises entropy only outside the core, and (iii) *global cosine*

*alignment* preserves overall direction. We use $\mathbb{E}_{b,h}[\cdot] = \frac{1}{BH} \sum_{b=1}^{B} \sum_{h=1}^{H} (\cdot)$ for averaging over batch and heads:

$$\mathcal{L}_{\text{core}} = \mathbb{E}_{b,h}\Big[\big\langle \tilde{a}_s(b, h, \cdot),\ M_{\text{core}}(b, h, \cdot) \big\rangle\Big], \tag{3}$$

$$\mathcal{L}_{\text{uni}} = \mathbb{E}_{b,h}\left[\sum_{n=1}^{N} p_{s,\text{nc}}(b, h, n)\ \log\ p_{s,\text{nc}}(b, h, n)\right], \quad p_{s,\text{nc}} = \frac{\tilde{a}_s \odot M_{\text{nc}}}{\|\tilde{a}_s \odot M_{\text{nc}}\|_1 + \varepsilon}, \tag{4}$$

$$\mathcal{L}_{\text{cos}} = \mathbb{E}_{b,h}\Big[1 - \cos\big(\tilde{a}_s(b, h, \cdot),\ \tilde{a}_t(b, h, \cdot)\big)\Big], \quad \cos(u, v) = \frac{\langle u, v \rangle}{\|u\|_2\ \|v\|_2}. \tag{5}$$

Here $\mathcal{L}_{\text{core}}$ measures how much $\tilde{a}_s$ still concentrates on the core region $M_{\text{core}}$. $\mathcal{L}_{\text{uni}}$ computes the (negative) entropy of the student attention restricted to the non-core mask $M_{\text{nc}}$; minimizing this term raises the entropy, thereby uniformizing attention outside the core and facilitating broader, more even exploration. $\mathcal{L}_{\text{cos}}$ preserve global directional consistency between student and teacher attention while still allowing local redistribution. Here $\mathbb{E}_{b,h}$ denotes averaging over batch and heads; $\odot$ denotes the Hadamard product. Overall, the introspective negative distillation loss is then

$$\mathcal{L}_{\text{IND}}\ =\ \alpha\,\mathcal{L}_{\text{core}}\ +\ \beta\,\mathcal{L}_{\text{uni}}\ +\ \gamma\,\mathcal{L}_{\text{cos}}. \tag{6}$$

The loss in Equation (6) promotes complementary redistribution: it steers the student away from the teacher's core peaks while keeping non-core attention diffuse and the global pattern aligned. $\alpha$, $\beta$, and $\gamma$ control the direction and strength of exploring novel cues.

**Phase I Loss Definition:**  The negative attention term $\mathcal{L}_{\text{IND}}$ primarily serves as an exploration prior and does not by itself guarantee class separability, we retain cross-entropy $\mathcal{L}_{\text{CEI}}$ as the sole supervision directly aligned with labels. $\mathcal{L}_{\text{CEI}}$ anchors the decision boundary and stabilizes optimization across tasks and it is given by:

$$\mathcal{L}_{\text{CE-I}}\ =\ \frac{1}{|\mathcal{D}_{\text{mix}}|} \sum_{(x,y) \in \mathcal{D}_{\text{mix}}} -\log p_\theta\big(y \mid x\big). \tag{7}$$

The cross-entropy term is applied to the same dataset $\mathcal{D}_{\text{mix}}$ as $\mathcal{L}_{\text{IND}}$. The two components are combined through a weighted sum in which the coefficient $\lambda_{\text{ind}}$ explicitly governs the influence of the term $\mathcal{L}_{\text{IND}}$. The complete Phase-I objective therefore reads:

$$\mathcal{L}_{\text{P-I}} = \mathcal{L}_{\text{CE-I}} + \lambda_{\text{ind}}\mathcal{L}_{\text{IND}}. \tag{8}$$

This design realizes a computational analogue of hippocampal *pattern separation*. After training, the student model $f_{\theta_{\text{init}}}$ assimilates the novel classification knowledge discovered through aggressive exploration, evolves into $f_{\theta_{\text{I}}}$, and will serve as hippocampal teacher in the subsequent phase.

### 3.2 PHASE II: CORTICAL CONSOLIDATION

Grounded in the two-phase memory theory, Phase II plays the role of systems consolidation: the information obtained in Phase I is frozen as a teacher $f_{\theta_{\text{I}}}$ and the most diagnostic attention patterns are transplanted into the long-term cortical backbone. This selective transfer preserves past performance while injecting genuinely novel cues(Foster & Wilson, 2006), yet practical challenges often hamper implementations. Therefore, HC2D employs two teachers to guide the student network in the second phase. The first teacher $f_\phi$ is the snapshot of the previous task; the second one $f_{\theta_{\text{I}}}$ is the hippocampal teacher that highlights the most discriminative cues discovered so far. For both teachers and the student we query the last block to extract CLS-to-patch attention vectors $a_{\text{pos}} = a_\phi$, $a_{\text{neg}} = a_{\theta_{\text{I}}}$, $a_s \in \mathbb{R}^{B \times H \times N}$. All attention extraction, normalization and layer-freezing setting are kept identical to those in Phase I. $f_\phi$ enforces full inheritance of prior-task attention, whereas the hippocampal teacher $f_{\theta_{\text{I}}}$ injects new class-discriminative cues without compromising the inherited attention. Unless otherwise stated, Phase II uses $\mathcal{D}_{\text{mix}}$, the same dataset as Phase I.

**Positive Distillation:** Here, the teacher $f_\phi$ is the frozen snapshot of the backbone at the end of task $t-1$. To directly inherit the teacher's spatial evidence, we perform an positive attention distillation on the CLS-to-patch attentions extracted from the last decoder block. Following the Phase I alignment and normalization protocol, we obtain the final attentions $\tilde{a}_s, \tilde{a}_{\text{pos}} \in \mathbb{R}^{B \times H \times N}$. Then we minimize the Kullback-Leibler divergence from teacher to student:

$$\mathcal{L}_{\text{POS}} = \mathbb{E}_{b,h}\left[\sum_{n=1}^{N} \tilde{a}_{\text{pos}}(b, h, n)\, \log \frac{\tilde{a}_{\text{pos}}(b, h, n)}{\tilde{a}_s(b, h, n)}\right]. \tag{9}$$

This loss (9) enforces full inheritance of the teacher's prior-task attention by the student. This unfiltered positive attention distillation provides a strong and stable continuity prior for Phase II by transferring, the frozen teacher's complete spatial distribution to the student. On the one hand, it acts as a stabilizer that anchors attention patterns and markedly suppresses forgetting; on the other, it complements the hippocampal teacher: while the hippocampal guidance encourages the discovery of novel cues, positive attention distillation supplies persistent spatial evidence that keeps exploration tethered to the existing semantic scaffold.

**Asymmetric Distillation:** Positive Attention Distillation forces the student to mimic the teacher on all tokens. In this situation, conventional knowledge distillation symmetrically aligns every teacher-student channel. During Positive Distillation, the student is already trained to match the teacher's full attention $f_\phi$ from the previous task. In practice, the teacher $f_{\theta_{\text{I}}}$ pushes the student to redistribute focus to novel cues, while teacher $f_\phi$ pulls it back to the teacher's entire map. If we simultaneously apply an unfiltered attention distillation from the hippocampal teacher $f_{\theta_{\text{I}}}$, the two supervision signals tend to pull the student toward different way, thereby creating a tug-of-war in optimization and degrading performance. To reconcile these extremes, we adopt an asymmetric distillation strategy. We therefore add an asymmetric distillation term that anchors the student only on the hippocampal teacher's most salient tokens (top-mass) and applies a one-sided penalty—enforcing sufficiency where the student under-attends, but not penalizing extra focus elsewhere. This removes the direct conflict with the positive distillation signal, preserves indispensable evidence on critical regions. Because all the updates are concentrated in the last Transformer block and the classifier head, the two signals act in a consistent direction at the top of the network, enabling a smooth fusion of attentions. Concretely, we follow the same alignment and token-normalization protocol as in Phase I, yielding attentions $\tilde{a}_{\text{neg}} \in \mathbb{R}^{B \times H \times N}$. Let $\sigma_{b,h}$ sort $\tilde{a}_{\text{neg}}(b, h, \cdot)$ in descending order and define the smallest prefix that covers a mass ratio $\rho \in (0, 1)$, and by Equation (1), we obtain $k_{\text{II}}^\star(b, h)$. The anchor mask is $M_{\text{anc}}(b, h, n) = \mathbb{I}\big(n \in \{\sigma_{b,h}(1), \ldots, \sigma_{b,h}(k_{\text{II}}^\star)\}\big)$, with $M_{\text{anc}} \in \{0,1\}^{B \times \hat{H} \times N}$ after head aggregation. We then form the anchored distributions:

$$p_{\text{neg}} = \frac{\tilde{a}_{\text{neg}} \odot M_{\text{anc}}}{\|\tilde{a}_{\text{neg}} \odot M_{\text{anc}}\|_1 + \varepsilon}, \qquad p_s = \frac{\tilde{a}_s \odot M_{\text{anc}}}{\|\tilde{a}_s \odot M_{\text{anc}}\|_1 + \varepsilon}. \tag{10}$$

The asymmetric loss penalizes the student only where it under-attends the teacher inside the anchor:

$$\mathcal{L}_{\text{NEG}} = \mathbb{E}_{b,h}\left[\sum_{n=1}^{N} \mathbb{I}\big[p_s(b, h, n) < p_{\text{neg}}(b, h, n)\big]\, p_{\text{neg}}(b, h, n)\, \log \frac{p_{\text{neg}}(b, h, n)}{p_s(b, h, n)}\right]. \tag{11}$$

This one-sided, top-mass anchoring preserves the hippocampal teacher's novel cues on the most salient tokens while never penalizing extra student focus outside the anchor. Because the constraint is both localized and directional, it does not compete with the main snapshot-guided branch ; instead, it complements it, enabling a seamless integration of new evidence with pulling the student back to the original full map. The primary distillation signal retains its primacy while asymmetric distillation injects the hippocampal cues smoothly into the final layer. This directional constraint preserves critical evidence, yet leaves ample capacity for exploratory feature discovery, which is crucial in continual learning scenarios where task relevance evolves over time.

**Phase II Loss Definition:** Beyond the distillation losses, we retain a conventional cross-entropy loss $\mathcal{L}_{\text{CE-II}}$ computed over the same mixed dataset $\mathcal{D}_{\text{mix}}$ used for Phase I. As defined in Eq. equa-

tion 7, we set $\mathcal{L}_{\text{CE-II}} = \mathcal{L}_{\text{CE-I}}$. To couple label-aligned supervision with attention-guided shaping, we blend cross-entropy with the attention-based regularizers to obtain the final loss:

$$\mathcal{L}_{\text{P-II}} = \mathcal{L}_{\text{CE-II}} + \lambda_{\text{neg}} \, \mathcal{L}_{\text{NEG}} + \lambda_{\text{pos}} \, \mathcal{L}_{\text{POS}}. \tag{12}$$

Equation 12 defines the Phase II loss: the cross-entropy $\mathcal{L}_{\text{CE-II}}$ anchors the decision boundary, while two complementary attention-distillation terms are superimposed-one preserves knowledge from the previous task, and the other injects new class-discriminative cues from the hippocampal teacher. The coefficients $\lambda_{\text{neg}}$, $\lambda_{\text{pos}}$ modulate the stability-plasticity trade-off.

## 4 EXPERIMENTS

In this section, we begin by clearly outlining experimental details and the specific training scenarios adopted. The implementation is built on the publicly available DyTox codebase(Douillard et al., 2021). In all experiments, we adopt the Distributed memory type; accordingly, all reported results and comparisons are computed under this implementation variant. To ensure rigorous and equitable comparisons, we meticulously maintain consistency in parameter settings and training strategies. Extensive experiments rigorously validate the effectiveness of the proposed HC2D framework in class-incremental settings and demonstrate its plug-and-play compatibility with existing methods. To systematically evaluate HC2D, we select the standard benchmarks CIFAR-100 (Krizhevsky & Hinton, 2009) and ImageNet-100 (Deng et al., 2009). Specifically, we employ task splits of $10 \times 10$, $20 \times 5$, and $50 \times 2$, corresponding to incremental learning scenarios consisting of 10 tasks with 10 new classes each, 20 tasks with 5 new classes each, and 50 tasks with 2 new classes each, respectively. To ensure a level playing field across all experiments, we employ the same ConViT backbone(d'Ascoli et al., 2021), thus guaranteeing identical backbone architectures across methods and ensuring experimental results remain unaffected by components external to the proposed framework. The combination of multiple loss terms and the complexity of the overall framework means numerous hyperparameters and some hyperparameters must still be set empirically, considering the characteristics of the training dataset and incremental task structure. For example, during incremental training over 10 phases on CIFAR-100, optimal balance between exploration and knowledge consolidation is achieved with $\rho = 0.8$, $q = 0.8$, $\alpha = 0.001$, $\beta = 0.01$, $\gamma = 0.1$. For ImageNet-100 or more steps, we adjust certain hyperparameters and learning rate schedules. For example, we increase the threshold ratio $\rho$ to 0.7 to accommodate higher image resolution and class complexity.

### 4.1 BASELINES COMPARISON

We evaluated four configurations to highlight the contribution of our framework. Base (vanilla) is a standard incremental learner without HC2D and trained with conventional objectives (cross-entropy on mixed batches) without the two-stage consolidation. Base w/ HC2D augments Base with HC2D framework, while keeping all other settings unchanged. DyTox is a strong existing baseline built on a ViT backbone(Dosovitskiy, 2020) with the task-token mechanism (Douillard et al., 2021). DyTox (w/o KL) disables the KL-based logit distillation in DyTox. DyTox (w/o KL) w/ HC2D is the method that seamlessly plugs HC2D into DyTox (w/o KL) to test HC2D's plug-and-play portability. We evaluated HC2D under two complementary comparisons: (i) Base (vanilla) vs. Base w/ HC2D, and (ii) DyTox (w/o KL) vs. DyTox (w/o KL) w/ HC2D. The former isolates the absolute gain of our framework over a standard baseline, whereas the latter probes plug-and-play portability to a strong continual-learning system that with the KL-based logit distillation disabled. As summarized in Table2, adding HC2D to the Base model lifts Last Acc (accuracy after the final task) by **+1.73** points and Avg Acc (mean accuracy over tasks) by **+2.96** points on CIFAR-100 dataset. Moreover, DyTox (w/o KL) w/ HC2D consistently outperforms DyTox (w/o KL) across all reported metrics. Figure2 provides an intuitive per-task view of accuracy, making the per-task changing trajectories and pairwise deltas immediately apparent. Across all tasks, result indicates that HC2D improves both early and late tasks rather than trading stability for plasticity.

To assess scalability and adaptability, we plug HC2D into the core task-token mechanism of DyTox and evaluate continual-learning performance under a fixed parameter count and memory footprint. As summarized in Table1, DyTox w/ HC2D consistently outperforms the DyTox baseline in both Avg Acc and Last Acc given identical model capacity, demonstrating strong practical utility. Across

Table 1: Comparison on CIFAR-100 under different class-incremental setups with the buffer size of 2,000. #P (number of parameters), average accuracy (Avg Acc), and last accuracy (Last Acc); "–" denotes unavailable entries. The table presents results comparing DyTox w/ HC2D against the baseline and prior methods.

| Methods | 10 steps | | | 20 steps | | | 50 steps | | |
|---|---|---|---|---|---|---|---|---|---|
| | #P | Avg Acc | Last Acc | #P | Avg Acc | Last Acc | #P | Avg Acc | Last Acc |
| ResNet18 Joint | 11.22 | – | 80.41 | 11.22 | – | 81.49 | 11.22 | – | 81.74 |
| Transf. Joint | 10.72 | – | 76.12 | 10.72 | – | 76.12 | 10.72 | – | 76.12 |
| iCaRL (Rebuffi et al., 2017) | 11.22 | $65.27_{\pm 1.02}$ | 50.74 | 11.22 | $61.20_{\pm 0.83}$ | 43.75 | 11.22 | $56.08_{\pm 0.83}$ | 36.62 |
| UCIR (Hou et al., 2019) | 11.22 | $58.66_{\pm 0.71}$ | 43.39 | 11.22 | $58.17_{\pm 0.30}$ | 40.63 | 11.22 | $56.86_{\pm 0.83}$ | 37.09 |
| BiC (Wu et al., 2019) | 11.22 | $68.80_{\pm 1.20}$ | 53.54 | 11.22 | $66.48_{\pm 0.32}$ | 47.02 | 11.22 | $62.09_{\pm 0.85}$ | 41.04 |
| WA (Zhao et al., 2020) | 11.22 | $69.46_{\pm 0.29}$ | 53.78 | 11.22 | $67.33_{\pm 0.15}$ | 47.31 | 11.22 | $64.32_{\pm 0.28}$ | 42.14 |
| PODNet (Douillard et al., 2020) | 11.22 | $58.03_{\pm 1.27}$ | 41.05 | 11.22 | $53.97_{\pm 0.85}$ | 35.02 | 11.22 | $51.19_{\pm 1.02}$ | 32.99 |
| DER w/o P (Yan et al., 2021) | 112.27 | $75.36_{\pm 0.36}$ | 65.22 | 224.55 | $74.09_{\pm 0.33}$ | 62.48 | 561.39 | $72.41_{\pm 0.36}$ | 59.08 |
| DyTox (Douillard et al., 2021) | 10.73 | 71.50 | 57.76 | 10.74 | 68.86 | 51.47 | 10.77 | 64.82 | 45.61 |
| DyTox w/ HC2D | 10.74 | $71.92_{\pm 0.36}$ | $60.38_{\pm 0.43}$ | 10.74 | $69.59_{\pm 0.28}$ | $53.74_{\pm 0.32}$ | 10.78 | $65.71_{\pm 0.41}$ | $48.11_{\pm 0.53}$ |

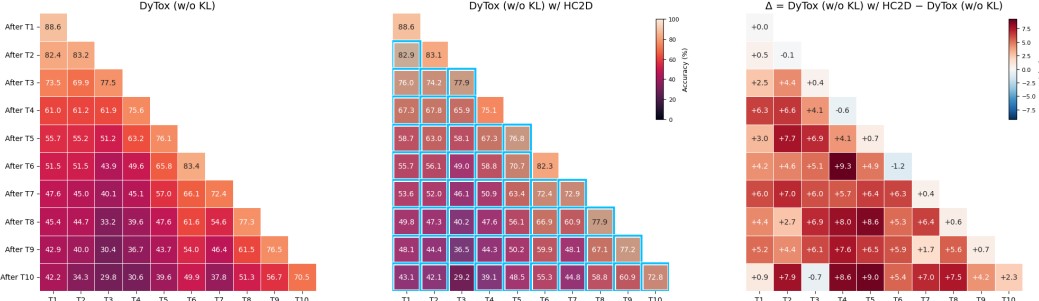

Figure 2: Per-task accuracy matrices for the baseline DyTox (w/o KL) (left) and DyTox (w/o KL) w/ HC2D (middle) under a 10-step class-incremental setting (rows: after step $k$, columns: task $T_i$). The light-blue boxes highlight tasks where DyTox (w/o KL) w/ HC2D exceeds the baseline. The right panel shows $\Delta =$, which indicating the gains attributable to HC2D, and the color intensity reflects magnitude of improvement or drop, and cell values denote percentage-point changes.

the experiments, HC2D delivers consistent improvements while honoring tight compute and memory budgets, validating both its effectiveness and practicality. In particular, HC2D mitigates forgetting and makes better use of previously acquired knowledge by reflectively refining decision details, leading to more balanced and efficient incremental learning.

## 4.2 ABLATION STUDY

We systematically conducted ablation experiments on the key components of our framework, carefully assessing the individual contributions of each module. Through these analyses, we aim to elucidate how each specific part influences overall performance and provide accurate insights. Starting from the complete model, we independently disabled Introspective Negative Distillation, Asymmetric Distillation, Positive Distillation, while keeping all other hyperparameters and implementation details fixed. For each configuration, we ran three independent trials on the 10-step class-incremental CIFAR-100 protocol with randomized task orders. Table2 summarizes the influence of each HC2D component across Forgetting (Fgt), Average Accuracy (Avg Acc) and Last Accuracy (Last Acc). Enabling all modules yields a 2.96 % increase in overall Avg Acc, confirming the effectiveness of the complete framework. By contrast, removing IND and asymmetric distillation decreases Avg Acc by 1.2 %. Without the introspection push away from teacher, the student completely freely explores new patterns, yet many of them overlap with old information, limiting diversity. Disabling asymmetric distillation reduces Avg Acc by 1.25 %. The two distillation loss terms may even trigger antagonistic gradients. And removing the positive distillation branch incurs a 1.28 % decline in average accuracy, indicating that although the model continues to explore new knowledge, it forfeits its capacity to preserve prior attention information. Collectively, these results demonstrate that only joint use of all three mechanisms closes the logical loop of HC2D, providing robust and consistent gains.

Table 2: Ablation study on CIFAR-100 with 10-step class-incremental setting. The table reports Last accuracy (Last Acc), Average accuracy (Avg Acc), and Forgetting(Fgt), and uses ✓/✗ to denote component presence or absence. "–" denotes a not-meaningful measurement.

| Positive Distillation | Asymmetric Distillation | Introspection Negative Distillation | Last Acc | Avg Acc |
|:---:|:---:|:---:|:---:|:---:|
| **HC2D** | | | **35.15** | **56.92** |
| ✗ | ✓ | ✓ | 32.87 | 55.64 |
| ✗ | ✗ | ✓ | 32.72 | 54.97 |
| ✓ | ✗ | ✗ | 33.95 | 55.72 |
| ✗ | ✓ | ✗ | – | – |
| ✓ | ✗ | ✓ | 33.11 | 55.67 |
| ✓ | ✓ | ✗ | – | – |
| **Baseline** | | | 33.42 | 53.96 |

## 4.3 COMPUTATIONAL AND MEMORY RESOURCES

In the continual learning scenario, computational and memory resources during both training and inference are critical metrics. HC2D is deliberately lightweight on both axes. During training, no auxiliary generators or intermediate representations are introduced by the framework. The memory module stores only raw or compressed images, whose capacity is bounded by a user-specified budget. Apart from retaining the frozen duplicate of the backbone as the hippocampal teacher at the end of Phase I, no further storage is required. Because HC2D's two-phase distillation operates only during training, the hippocampal teacher is discarded for inference, with all newly acquired knowledge merged back into the backbone. As reported in Table1, DyTox w/ HC2D matches Dy-Tox in parameter count, implying virtually no additional parameters in inference. Consequently, the deployed model closely mirrors the original backbone in footprint and latency, which means it preserves the same GPU memory usage during inference. Overall, HC2D introduces virtually no extra inference time or storage burden, yet delivers consistent performance gains a practical advantage for resource-constrained, long-horizon continual learning deployments.

## 5 CONCLUSION AND FUTURE WORK

This paper proposes Hippocampal-to-Cortical Two-Phase Distillation (HC2D), a continual-learning framework that leverages the intrinsic attention mechanism of Vision Transformer and the principles of introspection and asymmetric distillation to strike a balance between aggressive exploration and robust consolidation. Extensive experiments on CIFAR-100, ImageNet-100 and related benchmarks demonstrate that HC2D markedly reduces forgetting and maintains superior performance on long task sequences without incurring additional inference cost. During Pattern Separation, HC2D first leverages the ViT snapshot from the previous task and, via introspective negative distillation, mines high-information attention patterns complementary to the backbone, effectively completing the exploration of novel cues. During Cortical Consolidation, HC2D exercises token-level control via asymmetric distillation. This design avoids conflicts between two-phase attention-distillation branches and integrates the newly discovered cues. By coupling structural and algorithmic advantages, the method achieves a superior stability–plasticity trade-off. Ablation studies further verify that introspective negative distillation lays the discriminative foundation for aggressive exploration, and the two-phase framework and asymmetric distillation secure stable integration across the entire task stream, thereby validating the effectiveness of our biology-inspired design.

Despite the encouraging progress obtained so far, several critical avenues remain open for investigation. First, although the additional hippocampal teacher used during training is relatively lightweight, it still increases the memory footprint and introduces extra training time, potentially limiting deployment on resource-constrained edge devices. Future research may alleviate this burden through structural simplification, parameter sharing, and compression techniques. Second, the current framework exhibits notable sensitivity to hyperparameters configurations and depends on manually selected thresholds—such as mask ratios—whose optimal values may vary across tasks. Incorporating attention explainability analysis into the framework could supply principled guidance for adaptive hyperparameters tuning, thereby fostering more resilient continual-learning strategies.

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
