# OpenReview forum: "HC2D: Attention-Based Two-Phase Distillation for Transformer Continual Learning"
_ICLR.cc/2026/Conference — ICLR 2026 Conference Withdrawn Submission_

### Official Review · Reviewer_N5sz · 2025-10-25

**Soundness:** 2
**Presentation:** 3
**Contribution:** 1
**Rating:** 2
**Confidence:** 3

**Summary:**

The paper addresses catastrophic forgetting - the phenomenon where neural networks dramatically lose performance on previously learned tasks when trained on new ones. Existing approaches typically use local constraints around old decision boundaries, which limit the discovery of genuinely novel discriminative features. Therefore, this paper introduces HC2D (Hippocampal-to-Cortical Two-Phase Distillation), a novel framework for mitigating catastrophic forgetting in Vision Transformers during continual learning scenarios. The results show consistent improvements on CIFAR-100 and ImageNet-100 benchmarks.

**Strengths:**

1. Biological Inspiration: Successfully translates hippocampal-cortical memory theory into a practical machine learning framework, providing theoretical grounding.
2. Efficient inference: as mentioned in the paper, there are no additional parameters or computational overhead during inference - the hippocampal teacher is only used during training.

**Weaknesses:**

1. Baseline Selection: The paper compares HC2D against a baseline (DyTox) published in 2021, and disables KL-based distillation without clarifying the reason. It needs to compare with more recent and stronger baselines to validate the effectiveness of the proposed method.
2. Architecture Limitation: Specifically designed for Vision Transformers - unclear how well the approach would transfer to CNNs or other architectures.
3. Dataset limitation: only discuss the CIFAR-100 and ImageNet-100 datasets. Why just discuss the ImageNet-100 instead of the full ImageNet dataset? Because of high training overhead or bad performance for a larger dataset? Scalability to larger datasets or more complex scenarios remains unclear.
4. Hyperparameter Sensitivity: The authors acknowledge "notable sensitivity to hyperparameter configurations," which could limit practical deployment without extensive tuning.

**Questions:**

1. Why did you disable the KL-based distillation in the baseline DyTox? It is better to give more explanation.
2. Please provide performance comparisons with more recent and stronger baselines (e.g, SAFE[1] and PROOF[2]) or clarify the difficulty for performing the comparison.
3. How about the performance of the larger-scale dataset or other types of dataset tasks?
4. It is better to show and discuss the different settings of hyperparameters.

[1] SAFE: Slow and Fast Parameter-Efficient Tuning for Continual Learning with Pre-Trained Models (NeurIPS 2024)
[2] Learning without Forgetting for Vision-Language Models (TPAMI 2025)

---

### Official Review · Reviewer_Qycd · 2025-10-28

**Soundness:** 2
**Presentation:** 2
**Contribution:** 2
**Rating:** 2
**Confidence:** 4

**Summary:**

This paper proposes a continual learning framework for ViTs called HC2D, inspired by human memory systems. For each new task, it trains in two phases: A temporary Hippocampal Teacher model is trained.  It uses an Introspective Negative Distillation (IND) loss that "punishes" the model for using old attention patterns, forcing it to find new ones. Then, backbone model is trained. The authors claim this explore first, consolidate later strategy improves performance without adding inference cost.

**Strengths:**

1. The motivation is clear. Splitting training into exploration  and consolidation is a good analogy to human memory .

2. The complex two-phase process is only for training. The final model used for inference has no extra parameters or speed cost compared to the baseline, which is a practical advantage.

**Weaknesses:**

1. This method is impractical. It requires training twice for every new task. This doubles the total training cost. It needs very high training cost.

2. In Phase II, the model is guided by two conflicting teachers.  The loss from the old teacher forces it to copy the old attention map, while the loss from the new teacher forces it to copy a different, new attention map. It is a conflict and damage the learning.

3. The method adds so many new, sensitive hyperparameters, making the method almost hard to reproduce and use.

4. The entire complex distillation process only applies to the attention maps in the final Transformer block. All earlier layers are frozen during this process. This ignores rich features in shallower layers and is a major bottleneck.

5. The ablation only shows that removing parts of the model hurts performance.  This does not prove that this complex two-phase design is better than a simpler, single-phase method that just combines the same loss terms.

6. Introspective Negative Distillation is just a loss term that penalizes old attention peaks rather than the major techniques that its complex name sounds.

**Questions:**

See weakness part.

---

### Official Review · Reviewer_2Cob · 2025-10-30

**Soundness:** 1
**Presentation:** 2
**Contribution:** 2
**Rating:** 2
**Confidence:** 4

**Summary:**

This paper introduces Hippocampal-to-Cortical Two-Phase Distillation (HC2D), a continual learning strategy designed to address the critical challenge of balancing new knowledge exploration with the prevention of catastrophic forgetting. The proposed method employs a two-phase knowledge distillation process: Pattern Separation and Cortical Consolidation. In the initial Pattern Separation phase, HC2D promotes the exploration of new information through Introspective Negative Distillation (IND) while protecting previously acquired knowledge by freezing core features. The subsequent Cortical Consolidation phase focuses on integrating the newly learned information. This is achieved under the guidance of teacher model. Experimental results across various task-length settings demonstrate that HC2D outperforms baseline methods.

**Strengths:**

- The manuscript provides a clear and compelling motivation for the proposed HC2D algorithm.
- The concept of masking core features is interesting. It appears to strategically redirect the model's learning focus toward new information while protecting essential, previously learned knowledge.
- The use of two distinct distillations in the second phase thoughtfully balances the dual objectives of preventing knowledge drift (maintaining stability) and ensuring the effective acquisition of new knowledge (maintaining plasticity).

**Weaknesses:**

- My primary concern is that the proposed method relies on a rehearsal-based strategy, which inherently increases memory storage costs. More importantly, this approach deviates from the strict continual learning constraint, where access to previous data is typically prohibited. The authors should provide a clear justification for adopting this memory-based design over existing rehearsal-free approaches (e.g., L2P, DualPrompt, SDLoRA). To ensure a fair evaluation of the method’s effectiveness, an additional experiment without the memory buffer should be included.

- Another major issue is that the HC2D framework appears computationally expensive. During training, it requires forwarding each input through the ViT multiple times, which can be costly, especially for larger architectures. Moreover, the method involves cloning model parameters from recent states and includes a second training stage with both student and teacher models. This setup raises concerns about practical efficiency. The authors should provide a more objective evaluation, including metrics such as total training time and GFLOPs, compared to baseline methods during training.

- The paper claims to conduct experiments on ImageNet-100; however, the detailed results are missing from both the main text and the appendix. These results should be explicitly included, as **the current experimental evaluation is limited to a single dataset**.

- The results tables notably omit the forgetting rate metric. Reporting this measure is crucial, as it directly supports the claim that the Positive Distillation component mitigates catastrophic forgetting. Although the paper mentions the inclusion of forgetting analysis, no corresponding figures or results are provided.

- Another point of confusion is that Table 1 reports different accuracies for Transf. Joint across steps, whereas joint training should produce consistent results throughout, as all data are available simultaneously.

**Questions:**

Please refer to the Weaknesses section.

---

### Official Review · Reviewer_Cw8u · 2025-10-31

**Soundness:** 1
**Presentation:** 2
**Contribution:** 1
**Rating:** 2
**Confidence:** 4

**Summary:**

This paper proposes attention-based two-phase distillation for transformer continual learning. The 2 phases are patter separation and cortical consolidation. The paper proposes an interesting and unique idea. However, there are several fundamental issues that need to be addressed; please see the weaknesses.

**Strengths:**

(1). The paper has an interesting and unique idea.

**Weaknesses:**

Weakness:
(1). From the perspective of methodology, the proposed HC2D has no solid fundamental theory. I can not follow how the formulated loss improves the model performance.  Derivations and theoretical analysis will help the readers on how the proposed method is formulated and guaranteed to improve the trained model.

(2). Confusing term: what is teacher attention (a), novel-cue attention (b), asymmetric distillation processing (c), and fused attention (d). Are they feature-like vectors? Or functions? Or something else?

(3). Technical issue: The experiment is conducted only on CIFAR100 dataset, which is relatively an easy problem in CL. I can not find the result on ImageNet-100 as mentioned in section 4.

(4) Performance issue: In the 20-steps and 50-steps experiments, the HC2D makes only very small improvements, i.e,  < 1%.

(5). There is no catastrophic forgetting measurement and discussion about it, while CL is motivated by CF.

(6). Since the proposed method is for ViT-based CL, it should be applicable to any CL methods ViT backbone, including pre-trained ViT with parameter-efficient finetuning (Promp/Adapter/LoRA). However, the proposed method was only evaluated as a plugin for DyTox.

(7). No pseudo-code to explain in detail how the proposed method works.

**Questions:**

Please address the weaknesses.

---

### Note · Authors · 2026-01-14

I have read and agree with the venue's withdrawal policy on behalf of myself and my co-authors.